# Clinical and Prognostic Significance of the Eighth Edition Oral Cancer Staging System

**DOI:** 10.3390/cancers14194632

**Published:** 2022-09-23

**Authors:** Yasmin Ghantous, Aysar Nashef, David Sidransky, Murad Abdelraziq, Kutaiba Alkeesh, Shareef Araidy, Wayne Koch, Mariana Brait, Imad Abu El-Naaj

**Affiliations:** 1Department of Oral and Maxillofacial Surgery, Baruch Padeh Medical Center, Faculty of Medicine, Bar Ilan University, Ramat Gan 15208, Israel; 2Department of Oncology, The Sidney Kimmel Comprehensive Cancer Center, Johns Hopkins University School of Medicine, Baltimore, MD 21287, USA; 3Department of Otolaryngology and Head & Neck Surgery, Johns Hopkins University School of Medicine, Baltimore, MD 21205, USA

**Keywords:** oral carcinoma, depth of invasion, ENE, prognosis, adjuvant therapy

## Abstract

**Simple Summary:**

TNM staging of oral cancer is considered the cornerstone in treating and managing patients because it determines the need for adjuvant therapy. The eighth edition of the TNM staging system, when integrated into our clinical practice, highlighted the need for adjuvant therapy in a group of patients who were not recommended for treatment according to the previous TNM staging system. As adjuvant radio/chemotherapy has a significant effect on the patient’s quality of life, we planned and conducted this clinical study to evaluate the prognostic value of the current TNM staging system.

**Abstract:**

Objectives: The most notable changes in the eighth edition of the AJCC Cancer Staging System include incorporating the depth of invasion (DOI) into T staging and extranodal extension (ENE) into N staging. In this study, we retrospectively assessed the prognostic and clinical implications of the eighth TNM staging system. Materials and Methods: Patients with Oral Squamous Cell Carcinoma (OSCC) who were treated surgically between 2010 and 2017 were retrospectively reviewed. Tumors were first staged according to the seventh edition and restaged using the eighth edition. The prognostic value of the resultant upstaging was evaluated. Results: Integrating the DOI into the T classification resulted in the upstaging of 65 patients, whereas incorporating ENE into the N staging resulted in the upstaging of 18 patients *(p* < 0.001). Upstaging due to DOI integration had no significant effect on OS or DSS (*p* > 0.05). Conclusion: Our results demonstrate the importance of incorporating ENE into nodal staging and considering adjuvant therapy when ENE is present.

## 1. Introduction

Oral cancer is one of the most common cancers worldwide, with 90% of these being squamous cell carcinomas. Globally, head and neck cancers constitute 5.7 percent of cancer-related mortality [1]. Tobacco consumption and alcohol consumption remain the main etiological factors. HPV has occasionally been implicated in oral cancer development. Oral Squamous Cell Carcinoma (OSCC) is treated surgically. Therefore, sufficient pathological and histological data are available for prognostic evaluation and treatment planning. The factors affecting treatment choice are based on patient-related factors such as physical performance, daily activity, and overall health status. The primary site, location, size, proximity to the bone, and depth of infiltration are known factors that influence the surgical approach that will be undertaken. The OSCC staging system is divided into clinical and pathological stages. It is composed of three parameters: the size of the primary tumor (T), cervical lymph node status (N), and the presence of distant metastasis (M). The TNM staging system is the main component in determining treatment strategies and prognosis. However, some pathological and histological deficiencies, such as histological differentiation, lymphovascular involvement, depth of invasion, and extranodal extension [2,3,4,5], lead to incorrect risk stratification and insufficient treatment modalities. Therefore, an updated and more comprehensive TNM staging system was proposed in June 2018 based on several previously published scientific reports [2]. The main difference in the eighth edition of the Union for International Cancer was the inclusion of pathological features in the original TNM staging. Depth of invasion (DOI) and extranodal extension (ENE) were both incorporated into the T and N staging, respectively [2]. DOI has been shown to play a significant role in locoregional recurrence, spread, survival, and lymph node metastasis [3,4,5], and it was proposed to be an independent risk factor for treatment failure and recurrence. Hence, it is essential to include it in the new TNM staging system. Subsequently, in the eighth TNM edition, tumors that were formerly staged as T1 were upstaged to T2 if the DOI of the primary tumor was greater than 5 mm, and to T3 if the DOI was greater than 10 mm. Another major modification to the eighth edition of the AJCC staging system was the incorporation of ENE into N-staging. While the number, size, and location of nodular metastasis were already considered in the seventh AJCC edition, the presence of ENE is considered a poor prognostic factor based on several studies [6,7]. ENE upstaged the pathological N category by one level in the revised eighth edition criteria compared with the seventh edition. Clinically, overt ENE can upstage any clinical N into stage N3b.

This new staging system has consequences that may lead to different treatment strategies and combined treatment modalities, including adjuvant chemoradiotherapy. However, it is unclear whether this new approach will result in a better prognosis for affected patients. The purpose of this study was to retrospectively evaluate the prognostic and clinical performance of the eighth TMN staging system compared to the seventh edition.

## 2. Materials and Methods

Three hundred and three OSCC patients who were treated surgically between January 2010 and December 2017 (included) at the Department of Oral & Maxillofacial Surgery, Poriya Medical Center Israel, and at the Department of Otolaryngology and Head & Neck Surgery at the Johns Hopkins Hospital, Baltimore, MD, USA, were retrospectively reviewed. All patients were treated with curative intent by surgery, with the primary objective of achieving a microscopic clearance of 0.6 to 1.0 cm; adjuvant radiotherapy was administered according to cervical lymph node status. The inclusion criteria included the primary tumor of the oral cavity, resection of the primary tumor with oncological margins as described above, pathologically proven SCC as determined by a head and neck pathologist expert, and sufficient clinical and pathological data. Patients with an unknown primary tumor, history of head and neck cancer, and prior radiation therapy of the head and neck region were excluded. In addition, patients with microscopic margins less than 0.5 cm were excluded from the study (11 patients). Twenty-seven patients were excluded from the study due to insufficient pathological data regarding the depth of invasion. The patients were followed up for a minimum of 13 months. The data used in the current study was retrieved according to the approval of the local Institutional Review Board or other ethics committee at Poriya Medical Center (Approval code: 0060-22-POR) and at Johns Hopkins University (Approval: CR00037209/NA_00036235). Specific patient consent was waived due to the retrospective nature of this study.

Comprehensive clinical examinations, including head-and-neck computed tomography and/or MRI imaging, were reviewed retrospectively. The pathologic findings were assessed with particular attention to the depth of invasion, defined as the distance from the basal epithelial membrane to the deepest point of tumor cell infiltration; the epithelial surface was reconstructed in cases where the tumors were ulcerated or exophytic [8]. Based on these data, the tumors were first staged according to the 7th edition of the AJCC Cancer Staging System (2010) and re-staged using the 8th edition of the AJCC Staging System incorporating DOI and ENE. ENE was defined as the spread of tumor cells >2 mm across the capsule. The current study focused mainly on tumors that were upstaged to T-stage T2 and T3 (based on the depth of invasion) and N-stage tumors based on the presence of ENE [5].

### Statistical Analysis

Statistical analysis was performed using SPSS version 22.0 (United States Software Announcement 213-309 IBM Corp. The clinical endpoints of interest were overall survival (OS) and disease-specific survival (DDS). Survival time was calculated from the date of surgery to the last follow-up date. Univariate comparisons between groups were performed using the log-rank test. For DSS, patients who died of causes other than OSCC were censored at the time of death. A significance level of 5% was used for all statistical analyses. For continuous variables (depth of invasion, age), univariate analysis was performed by a comparison of means using the t-test or Mann–Whitney U test, as appropriate. Kaplan–Meier survival curves were constructed for categorical variables, including the pathological T stage and overall TNM staging, and significance was calculated using the log-rank test (Cox–Mantel). All clinicopathological factors that were predictive of survival (including depth of invasion and tumor subsite) in univariate analysis were entered into a Cox regression model to establish the independent predictors of survival.

In summary, the prognostic value of the studied variants (DOI, tumor subsite, ENE, and clinical variants) was studied in several ways: (1) through their statistical significance in multivariate analyses; (2) the Cox’s proportional hazard regression model, which was used to calculate the separation between the survival curves in each TNM system, while the T1 category was used as the comparator; and (3) finally, compared to multivariate models with and without the covariate of interest, using the two log-likelihood ratio test to determine whether the model (the TNM staging system) fit was significantly improved. The *p* significance was set at *p* < 0.05.

## 3. Results

### 3.1. Clinical and Pathological Results

#### 3.1.1. Patients Clinicopathologic Data

Overall, 265 patients were included in the current study (160 patients were treated at the Department of Otolaryngology and Head & Neck Surgery at the Johns Hopkins Hospital, Baltimore, MD, USA; 105 patients were treated at the Department of Oral & Maxillofacial Surgery, Poriya Medical Center Israel). Two hundred and two tumors were considered as stage I and II, and 63 as stage III, according to the seventh TNM staging system (the demographic and clinicopathological data of the study cohort are summarized in Table 1).

The study cohort included 94 females (35%) and 171 males (65%), with a median age of 63 years (and a range of 22–90 years). Of these patients, 127 (48%) reported tobacco use (current and/or previous) and 77 (29%) reported alcohol consumption. Most of the primary tumors were located in the oral tongue (62%), followed by the floor of the mouth, lower alveoli, and buccal mucosa (23%, 8%, and 3%, respectively). One hundred and sixty-five primary tumors were staged as T1 in the final pathological staging system (seventh edition). Overall, 60% of patients underwent elective neck dissection. Seventy-five percent of the patients underwent level I–IV neck dissection, while the remaining twenty-five percent underwent level I–III neck dissection, as described by the American Head and Neck Society. Almost half of the patients with stages I and II tumors (52%) underwent elective neck dissection, which was pathologically staged as N0. Almost 85% of the patients with stage III disease (seventh edition) underwent elective neck dissection, as described by the American Head and Neck Society [9].

The average depth of invasion (DOI) of all patients included in the study, as determined by pathological analysis, was 11.12 mm (range: 1–22.2 mm). The average DOI among stage I and stage II tumors (seventh edition) was 5.51 mm (median: 4.75, range: 1–21 mm). Treatment modalities included surgery alone in almost 75% of the patients (198 patients), whereas 15% were treated with primary surgery followed by radiotherapy. The follow-up period ranged from 13 to 431 months (median: 76 months).

#### 3.1.2. Upstaged Patients According to the 8th AJCC Edition of the Study Group

Of the 265 patients, the eighth AJCC edition resulted in upstaging in 83 (31%). The group included 23 females and 60 males (average age: 56.5 years). The tumors in these cases were mainly located in the oral tongue, followed by the floor of the mouth (42 and 11 patients, respectively). Seventeen (8.5%) out of the two hundred and two patients who were initially considered to have early-stage disease (stage I or II), according to the seventh edition, were upstaged to advanced disease (stage III) using the eighth edition of the AJCC staging system, and eighteen stage III patients (seventh edition) were upstaged to stage IVa disease. Integrating DOI into the primary tumor (T) classification, according to the eighth TNM staging system, resulted in the upstaging of 65 patients. Of these, 48 patients were considered to have a T-stage 1 disease according to the seventh edition but were upstaged to a T-stage 2 disease according to the eighth AJCC edition. Additionally, 5 patients were staged as T-stage 1 (seventh edition) and were upstaged to T-stage 3, and 12 T-stage 2 patients were upstaged to a T-stage 3 disease according to the eighth edition. The pathological nodal staging was upstaged in 18 N-stage 3 patients (seventh edition) due to the presence of ENE, while ENE was defined as tumor cells spreading more than 2 mm from the node capsule. Only two patients were upstaged in both T and N staging (upstaging from stage II to stage III).

#### 3.1.3. The OS and Five Years DSS Were Analyzed between the Sub-Groups

Group (1): The OS of patients staged as pT1 and pT2 N0M0 by the seventh edition of the AJCC staging manual was 92% (CI 89–95%) compared with the 5-year OS of all the patients who were upstaged to pT2 and pT3 N0M0, which was 95% (CI, 86–92%) according to the eighth edition of the AJCC staging system (*p*-value = 0.07) (Figure 1).

Group (2): The OS of patients upstaged to pT3 (from pT1, pT2N0M0) was 94% (CI: 89–96%), compared with 92% (CI:89–97%) in patients who remained in pT1 and pT2 N0M0 (*p*-value = 0.082, Figure 2).

Group (3): The 5-year OS of patients staged as stage III according to the seventh edition of the AJCC staging manual and upstaged to stage IVa according to the N category of the eighth edition, compared to patients that remained at stage III was 78% and 89%, respectively (*p*-value < 0.005) (Figure 3).

In addition, only one patient in the stage I group (seventh edition) had local recurrence at the 14-month follow-up, and two patients in the stage II group had locoregional recurrence.

#### 3.1.4. Multivariate Analysis

Multivariate analysis revealed several factors in addition to the seventh TNM staging, which affected both OS and disease-free survival. The pT staging and the presence of ENE were especially related to worsened OS and disease-specific survival as well (*p*-value *=* 0.043 and <0.001, respectively).

Moreover, both the tumor subsite and alcohol consumption significantly affected the OS of the patients. Primary tumors located in the tongue and buccal mucosa were found to have a significantly lower OS than the remaining tumor subsites (68% and 71%, respectively, vs. 86%, *p*-value =0.042 and 0.034) (Table 2).

In addition, patients with a history of alcohol consumption had worse OS than those who did not report alcohol consumption (82% vs. 92%), regardless of sex or age (*p*-value = 0.02).

## 4. Discussion

Oral squamous cell carcinoma is one of the most commonly diagnosed cancers worldwide, especially in central Europe and Asia, and still accounts for almost 145,000 deaths per year [10]. The prognosis of OSCC has remained stable for the last four decades, with approximately 85% for stage I and II diseases and nearly 45% for more advanced conditions. However, the prognosis drops by 50% when metastatic disease is found in the cervical lymph nodes [10]. In addition, prognosis strongly depends on the tumor subsite and individual habits, such as smoking and alcohol consumption [11,12].

In 1977, the American Joint Committee on Cancer published the Premier Cancer Staging Manual, which remained unchanged. It was officially published in 2009 as the seventh edition of the AJCC staging system for oral cancer. Tumor staging is essential because it provides solid guidelines for the treatment and management of patients. The seventh AJCC system for oral cancer incorporates the tumor size, presence, location, and size of cervical lymph node metastasis and the presence of distal systemic metastasis. However, it does not consider pathological findings, such as the depth of invasion of the primary tumor or tumor extension with regard to cervical lymph node metastasis. These two pathological parameters have been found in several studies to play a significant role in determining the prognosis of patients and their ability to achieve disease control [11,12,13,14]. These elements were mainly related to the presence of lymph node metastasis when the DOI was >4 mm, regardless of the anatomical subsite. Thus, neck dissection is indicated in tumors with a DOI greater than 4 mm [14,15]. DOI was integrated into the AJCC staging of several cancers, such as cutaneous SCC, esophagus, melanoma, stomach, and rectum, and the invasion of a specific anatomical layer was considered a critical feature in determining the staging of previous cancers [9]. Notwithstanding, several studies failed to find a direct relationship between DOI and prognosis [15,16,17,18]; however, they did find a strong correlation between DOI and the risk of nodal metastasis, especially in early-stage tumors. In addition, a study published by Kano et al. found no significant difference in the DSS between T1 and T3 patients (in the seventh edition compared to the eighth edition) [19]. The authors concluded that tumor T staging for T1 and T3 patients was well-addressed in the seventh edition, as most T1 and T3 tumors had a DOI less than 5 mm or >10 mm, respectively [16]. Furthermore, DSS was not significantly different between T1 and T2 patients based on the seventh and eighth editions [19]. Similarly, our study also failed to demonstrate a significant difference in DSS and OS for stage I and II patients (seventh edition), who were upstaged to a more advanced disease based on the eighth edition. Moreover, it is well-known that DOI correlates with the presence of cervical lymph node metastasis, and many studies have shown that a tumor infiltration greater than 10 mm harbors a high probability of lymph node metastasis [16,18,20,21].

Different anatomical barriers, such as extrinsic tongue muscles, gradually vanish with a depth of invasion more significant than 10 mm from the mucosal surface, which may enable tumor cells to migrate more easily to adjacent lymphovascular vessels. Because the presence of lymph node metastasis is the most important prognostic factor in OSCC, incorporating this factor into the TNM staging system is very important. However, according to the literature, the optimal cut-off DOI (which indicates the need for elective neck dissection) varies from 3 mm to 10 mm, and it depends substantially on the tumor subsite [20,21,22]. Thus, determining a unified DOI threshold for upstaging T1 to T2 is crucial. This definition could result in the added morbidity of neck dissection for those who are upstaged to T2 and potentially spare T1 patients who could still be at risk of developing occult metastatic lymph nodes and locoregional recurrences. Therefore, optimal T staging should not be based solely on two parameters (tumor size and DOI). It may be crucial to consider several factors, such as the pattern of invasion, differentiation, histological grading, and the tumor subsite.

By contrast, the present study demonstrates the importance of integrating ENE when assessing cervical lymph node metastasis for N staging. The presence of ENE significantly reduced the OS and DSS of affected patients when comparing their survival at stage III, according to the seventh edition of the AJCC staging manual, to their revised stage IVa status according to the eighth edition N category. ENE was found to be a significant risk factor for regional recurrences [23,24]. ENE is pathologically defined by the spreading of metastatic tumor cells through the fibrous capsule and into the surrounding connective tissue; a minor ENE is determined when the tumor cell spreading is within 2 mm from the node capsule, while a central ENE is defined when the tumor spreads more than 2 mm microscopely, or when the ENE is grossly seen in the surgical field [25]. Furthermore, the correlation between ENE, locoregional recurrence and distant metastasis has been extensively studied, and a direct relationship has been found between these parameters [6,26]. ENE was also associated with worse DSS, especially in HPV-negative patients [27]. Thus, the presence of ENE necessitates the acceleration of adjuvant therapy at every OSCC stage.

## 5. Conclusions

Our study demonstrated the importance of incorporating ENE into N staging and the need for adjuvant therapy whenever ENE is present. However, DOI was not an independent factor for a worse prognosis and thus did not support the addition of adjuvant treatment in our study. T staging should include other histological and clinical characteristics, including the tumor subsite and pattern of invasion. Notwithstanding, DOI is recommended as a relative indication for elective neck dissection in accordance with the tumor subsite and diameter.

## Figures and Tables

**Figure 1 cancers-14-04632-f001:**
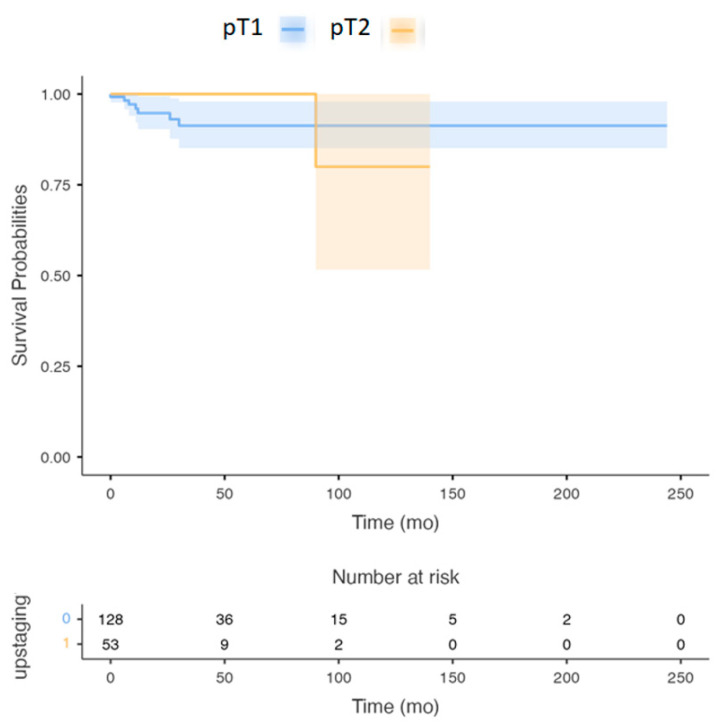
Survival Curve: Overall survival of pT1 and pT2 patients with the seventh AJCC staging system versus eighth AJCC staging system. (*p*-value > 0.05).

**Figure 2 cancers-14-04632-f002:**
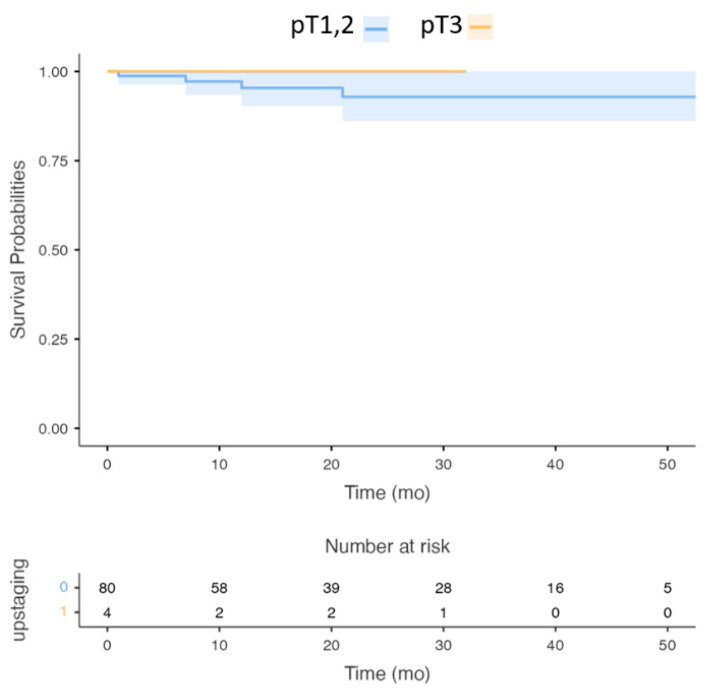
Survival Curve: The 5-year DSS of patients upstaged to pT3 (eighth AJCC staging system) from pT1, pT2N0M0 (seventh AJCC staging system). (*p*-value > 0.05).

**Figure 3 cancers-14-04632-f003:**
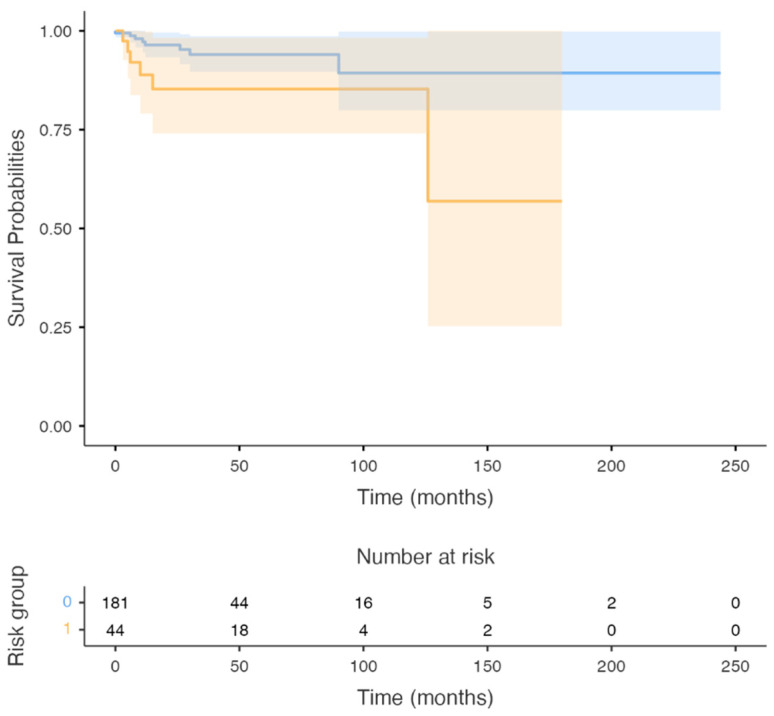
Survival Curve: The survival of patients staged as stage III by the seventh edition of the AJCC staging manual and who were upstaged to stage IVa according to the N category of the eighth edition, compared to patients that remained at stage III. (*p*-value < 0.005).

**Table 1 cancers-14-04632-t001:** Univariate analysis of the study cohort.

Variable	Num. (%)	*p* Value (*t*-Test)
Sex:		0.08
female	94	
male	171	
Mean age at diagnosis	63 (range 22–90)	0.3
Tobacco consumption	127 (48%)	0.06
Alcohol consumption	77 (29%)	0.02
pT staging 7th edition:		
T1	165	
T2	97	
T3	3	
T4	-	
pT staging 8th edition:		
T1	107	
T2	133	
T3	20	
T4	-	
Pathological N staging 7th edition:		
N0	204	
N1	60	
N2a	-	
N2b	-	
N3	-	
Pathological N staging 8th edition:		
N0	205	
N1	42	
N2a	18	
N2b	-	
N3	-	
Tumor sub-site:		
Oral tongue	164 (62%)	
Floor of mouth	60 (23%)	
Buccal mucosa	21 (8%)	
Lower alveolus	9 (3%)	
Upper alveolus	7 (3%)	
Retro-molar trigone	4 (1%)	
Neck Dissection	159 (60%)	0.37
Stage I	64	
Stage II	41	
Stage III	53	
Level I-III	39	
Level IV	120	
Extra-capsular extension	18	
TNM staging 7th edition:		
Stage I	140	
Stage II	62	
Stage III	63	
Stage IVa	-	
Stage IVb	-	
TNM staging 8th edition:		
Stage I	87	
Stage II	98	
Stage III	62	
Stage IVa	18	
Stage IVb	-	

**Table 2 cancers-14-04632-t002:** Multivariate analysis of the study cohort, pT1 was used as a reference.

Variable	Disease-Specific Survival *p* Value	HR CI 95%	Overall Survival *p* Value	HR CI 95%
pT staging 7th edition: *(Reference T1)*				
T1	-	-	-	-
T2	0.043	2.64 (1.19–5.83)	0.036	2.02 (1.09–3.37)
T3	0.67	0.5 (0.03–501)	0.6	0.85 (0.03–19.8)
T4	-	-	-	-
pT staging 8th edition: *(Reference T1)*				
T1	-	-	-	-
T2	0.023	2.1 (1.3–4.1)	0.03	2.5 (1.3–5.2)
T3	0.43	0.49 (0.65–501)	0.56	0.84 (0.02–24.6)
T4	-	-	-	-
Pathological N staging 7th edition: *(Reference T1)*				
N0	0.37	0.89 (0.92–1.04)	0.38	0.85 (0.26–2.57)
N1	<0.001	4.08 (1.96–8.48)	<0.001	3.83 (2.12–6.94)
N2a	-	-	-	-
N2b	-	-	-	-
N3	-	-	-	-
Pathological N staging 8th edition: *(Reference T1)*				
N0	0.49	0.26 (0.04–1.65)	0.12	0.42 (0.14–1.26)
N2a	<0.0001	5.97 (2.84–9.54)	<0.0001	5.1 (2.91–8.95)
N2b	0.04	3.6 (1.9–8.95)	0.032	4.55 (3.1–9.32)
N3	-	-	-	-
Tumor subsite: *(Reference T1)*				
Oral tongue	0.056	1.76 (0.49–2.3)	0.042	1.8 (1.1–3.21)
Floor of mouth	0.32	1.3 (0.01–4.5)	0.41	2.1 (0.3–6.67)
Buccal mucosa	0.07	2.3 (0.1–3.5)	0.034	1.98 (0.3–3.7)
Lower alveolus	0.9	0.95 (0.2–4.6)	0.85	1.3 (0.5–3.8)
Upper alveolus	0.57	3.1 (0.6–7.1)	0.8	3.6 (0.7–4.1)
Retro-molar trigone	0.87	2.01 (0.5–8.9)	0.69	1.9 (0.3–8.1)
Neck Dissection *(Reference T1)*				
Stage I	0.89	0.91 (0.17–3.62)	0.77	0.85 (0.26–2.57)
Stage II	0.8	1.3 (0.7–2.1)	0.85	1.6 (0.4–3.1)
Stage III	0.12	1.1 (0.1–3.7)	0.3	1.8 (0.09–3.5)
Extra-capsular extension	<0.001	4.39 (1.76–8.32)	<0.001	5.97 (2.84–9.91)
TNM staging 7th edition:				
Stage I	-	-	-	-
Stage II	0.65	2.1 (0.3–4.1)	0.54	2.3 (0.9–3.6)
Stage III	0.047	1.203 (1.044–2.094)	0.042	1.12 (1.001–3.01)
Stage IVa	0.03	1.2 (1.1–3.4)	0.035	1.5 (1.1–4.2)
Stage IVb	-	-	-	-
TNM staging 8th edition:				
Stage I	0.45	1.7 (0.6–4.2)	0.44	0.82 (0.02–24.6)
Stage II	0.2	1.17 (0.65–4.66)	0.12	0.98 (0.94–1.1)
Stage III	0.056	0.18 (0.0–1886)	0.084	0.3 (0.02–24.6)
Stage IVa	-	-	-	-
Stage IVb	-	-	-	-

## Data Availability

All the data of the study cohort can be reached through the first and second aouthors.

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
