# Peer review of "Clinical and Prognostic Significance of the Eighth Edition Oral Cancer Staging System"

_cancers, 2022, doi:10.3390/cancers14194632_

Round 1
Reviewer 1 Report (Previous Reviewer 2)
The revision is satisfactory.
This manuscript is a resubmission of an earlier submission. The following is a list of the peer review reports and author responses from that submission.
Round 1
Reviewer 1 Report
This retrospective study assessed the implications of 8th TNM staging system on the treatment and prognosis of oral cancer patients. The authors report that integration of DOI into the T-classification upstaged to 65 pts, whereas, inclusion of ENE in the N-classification upstaged 18 patients. However, tumor upstaging due to DOI inclusion had no significant effect on OS or DSS. But tumor staging due to ENE inclusion had a significant effect on the survival rates.
Did the authors follow-up on patients who initially had a lower TNM stage (per 7th edition) and received surgery only, and later according to 8th edition were upstaged. Did any of them have any recurrences or distant metastasis, or are deceased because of the disease?
My specific comments are on the attached pdf document.
Thanks

Author Response
Response to reviewer #1:
First I would like to thank you for your valuable comments, please see below the response for your comments:
- The introduction section was justified according to your specific comments
- The patients and methods section was edited according to your specific comments
- The result sections:
- The study included 265 patients, after the exclusion of 38 patients that did not meet the inclusion criteria. The first line in the result section was modified to fit your comment.
- 60% of all patients underwent neck dissection, this was added in the beginning of the paragraph.
- Tumor subsite and alcohol consumption significantly affected the OS of the patients. (Was added to the text)
- All other issues were addressed, including references.
Regarding your question, did the authors follow up on patients who initially had a lower TNM stage (per 7th edition) and received surgery only, and later according to the 8th edition were upstaged? Did any of them have any recurrences or distant metastasis, or are deceased because of the disease?
Only one patient in the Stage I group (according to the 7th edition) had local recurrence in 14 months of follow-up, and 2 patients in the Stage II group (locoregional). This was added to the result section.
Best Regards,
Dr. Yasmin Ghantous
Padeh Poriya Medical Center
Tiberires, Israel
Reviewer 2 Report
Several reports in the literature have tested the prognostic value of AJCC 8th edition new TNM staging for oral cancers with some more focused on the recurring fact that all oral tumors are not homogeneous and therefore should not be lumped together. In fact there are a number dedicated to the most common oral SCC, mobile tongue SCC. I believe such focused studies are better than those who lump all oral SCC together.
Do I feel this report is offering any new information or guidance on management? I am not sure.
Here are my few observations and queries reading through this manuscript.
1. The paper suffers from a lot of clerical errors with misplaced words and letter, wrong spellings and factual statements without references. In general that does not make for smooth reading of the manuscript. I also have issues with the way the results were presented.
2. References should be provided, for example, for the factual statements particularly in the “Introduction” section (page 2), page 3 (lines 102-103), page 4 (lines 144-45), page 7 (lines 208-10 and 211-212.
3. In the survival analysis, how was it possible to use T4 category as comparator when no patient had T4 lesions (page 3, lines 122-24)? This is just confusing to me.
4. The established clinical and treatment guidelines is that any nodal involvement in OSCC should usually have adjuvant therapy in addition to surgery. The authors did not make clear what other adjuvant therapy is being advocated upon discovery of ENE. I am not familiar with even weak evidence in support of not giving escalated management methods to any patient with ENE even in the period before the 8th edition of AJCC staging was published.
5. I will advise the authors to use standard methods of reporting survival analysis: hazard ratios and confidence intervals, rather than percentages.
6. The authors should label the figures (Kaplan-Meier curves) appropriately as pT1 and pT2 instead of upstaging 0 and 1. The same should apply to the N stage (stage III to IVa) to make things clear to any reader. The P value for the Kaplan Meier curves should also be stated in their figures. The diagrams are not clearly explanatory in their current form.
7. What does “controlling patients” (page 7, line 216) mean? If it is referring to following them up, is that still not part of the patient management?
8. The statement on page 8 about average upstaging DOI being 8.7 mm (lines 238-240) is confusing. The authors are advised to make it clearer.
9. The last lines in the conclusion section do not appear to be supported by anything derived from this work (lines 276-278), except the authors have other data not presented in the study.
In summary, I am not convinced about the need for this study.
Author Response
Respond to reviewer #2
First, I would like to thank you for your valuable comments.
Please see the below response to your comments.
- the manuscript was edited for spelling and errors.
- References were added accordingly.
- The analysis was edited, and the comparison to T4 was deleted.
- We agree with your comment that ENE patients always receive adjuvant therapy, regardless of TNM staging timing. This is emphasized in the discussion section, lines: 273-275.
- This was added and detailed in Table 1.
- the figures were edited according to your comments.
- yes, the word controlling was deleted.
- We meant that the DOI in our paper is less than the average needed for upstaging; this was added to the sentence.
- it is a recommendation based on our study and other studies. The word recommended was added.
Best regards,
Best Regards,
Dr. Yasmin Ghantous
Padeh Poriya Medical Center
Tiberires, Israel
Reviewer 3 Report
Thank you for inviting me to review the manuscript entitled “Clinical and Prognostic Significance of the Eighth Edition Oral Cancer Staging System”. In this manuscript, the authors investigated the Clinical and Prognostic Significance of the Eighth Edition Oral Cancer Staging System and evaluated the difference between 7th AJCC system and 8th AJCC system in oral cancer.
This is an interesting manuscript about the biologic role and diagnostic value of 8th AJCC system in OSCC. The authors provided well-conducted statistical analysis. I think that this manuscript will be a useful reference for the field. Several minor comments are listed below that authors need to clarify them.
1. In the Result section, adding a subhead would be better for the reader to understand. In the manuscript, only the numbers are written as 3.1.1, 3.1.2, etc. (Page 3 126 lines, Page 5 148 lines, Page 5 165 lines, Page 5 178 lines)
2. I think it would be better to change "3.3. Figure" in line 188 of page 2.5 to another subheading that can better explain the results of the experiment.
3. In Table 1, are the numbers written in the "Disease-specific survival" and "Overall Survival" columns p-value? If so, it would be good to specify that it is p value.
Author Response
Response to reviewer #3:
First I would like to thank you for your valuable comments, please see below the response for your comments:
- subheading was added
- the subheading was changed to Survival curves of the study groups (Kaplan Meir).
- Yes, it was added to the table
Best Regards,
Dr. Yasmin Ghantous
Padeh Poriya Medical Center
Tiberires, Israel
Round 2
Reviewer 1 Report
The content of the revised manuscript has improved the quality of the article. I suggest the authors thoroughly read the manuscript for improving the English language. Some grammatical mistakes are noted.
Good luck,
Author Response
Review #2
Thank you so much for your comment; the paper was sent to a professional English editor.
Best Regards

Reviewer 2 Report
Re-reviewing this case, it is my opinion that the authors have failed to adequately address the issue that were raised by me. It is more likely that this is as a result of the way the response was made. The best way to respond to a reviewer is to quote the statement he made, respond to it and indicate clearly where your response can be found. The authors did not do this.
I also discovered some new queries reading through the revised manuscript.
PREVIOUS ISSUES (relating to authors’ responses):
1. The English language still requires improvement.
2. The authors have thankfully provided many references, please see item 9 and new issues for more needed.
3. Deleting T4 is not enough. What was done in its place or what is now the new reference?
4. Adequate
5. Adequate
6. Adequate
7. Adequate.
8. This explanation about DOI is still confusing. What is the point in taking an average? >5mm depth converts T1 to T2 and >10mm converts T2 to T3. Upgrading from T1 to T2 should be taken separately from upgrading from T2 to T3. It appears the author lumped up both and came to an average of <10 mm. It really does not make much sense to conclude that that the patients here fared better because their average was <10 mm. Comparison can only be made between one upgrading to the next and not across all upgrading (by taking the average). I suggest the argument in lines 252-258 should be removed, unless the authors can better articulate what they are trying to say.
9. If the recommendation is by other studies, those studies should be referenced.
NEW ISSUES
In the Introduction section,
1. TNM staging is composed of 3 parameters and not 3 “main” parameters (line 49), and what is meant by “TNM staging systemis world” (line 51)?
2. Reference(s) should be provided for lines 52-53. What are the pathological and histological deficiencies noted?
In the Methods section,
1. The authors should state clearly the number of cases that either centers contributed to the total cases.
2. Any sentence that begins with a number should be written out in English (so 303 in line 77 should be written out clearly).
3. In the statistical analysis, lines 104 and 105 should be rephrased as “The clinical endpoints of interest were overall survival and disease –specific survival…”
In the Results section,
1. The new information from lines 135-140 should be included in Table 1 along with their exact numbers.
2. In Table 1, what is the purpose of the last column as it appears there is no mention of this either in the text of this section or in the Discussion section.
3. For disease-specific survival relating to tumor subsite, the stated confidence interval is not compatible with a p value of <.05 because the minimum value is less than 1. Except for p values less than 0.001, the authors should quote the specific p-values. They should avoid using < or >0.05.
4. Why were the DSS and OS of categories in TNM staging of 7th and 8th not evaluated ?
5. The whole paragraph of lines 145 to 149 is not clear (it should be properly rephrased). In addition it is not clear how the author got 85% of patients mentioned in line 147. There is no way to look at this and arrive at this percentage whether 198/202 or 198/265. The authors appeared to have muddled up the results here.
6. In the subsections 3.1.3, 3.1.4, the actual p values should be quoted.
7. In line 185, what is “worsened prognosis”? Is it OS or DSS? In result section, the authors should be specific about what is being reported or otherwise define any term being used.
8. Just to assert again, survival statistics including univariate and particularly the multivariate analysis, should be shown in the format of hazard ratios and 95% confidence intervals which has not been done by the authors here and a separate table should be created for the multivariate analysis.
In the Discussion section,
1. Lines 242-246 is confusing, on one hand there was the talk about “several papers” and then “a study by Kano and co-workers” using the same reference.
Author Response
Response to Reviewer #2
Thank you for your comments; please see attached a detailed response to your comments
Best Regards

Round 3
Reviewer 2 Report
The authors have made significant changes. Their effort should be commended. However, there are still outstanding issues.
1. I still have reservations about lines 255-257. It still appears completely wrong even with further explanation by the authors. Upstaging through DOI is not by increments of 10 mm. It is by increments of 5 mm (T1 to T2 to T3). This is why it sound so ridiculous that the authors are claiming that in upstaging if the average DOI is < 10 mm, then it is an explanation for it not being related to survival. That is not how to assess upstaging. It is not a score to be averaged. So taking an average is meaningless. How many studies have they seen taking averages of DOI cutting across all tumor stages? The reason given does not explain why their DOI had no effect on survival. It can only stand on its own if other studies where DOI had effect had their own average DOI stated, and it was more than 10 mm. The authors should provide references for that. The demand remains the same. They should delete this explanation that appears to make no meaning or provide the evidence from other studies employing the same argument or look for a more reasonable explanation.
2. For the multivariate analysis, since Cox regression model (which I understood as Cox proportional hazard model) was used, if one of the categories is used as comparator why were the values (HR, 95% CI, and P values) of the other categories not stated? This is referring to pT staging, pN staging, tumor subsite, neck dissection and TNM staging.
Author Response
Dear Reviewer
Thank you for your valuable comments; your comments were addressed in the paper.
please see attached a detailed respond (point by point) for your comments.
Best Regards,
Dr Yasmin Ghantous
